# Folate-Targeted Monodisperse PEG-Based Conjugates Made by Chemo-Enzymatic Methods for Cancer Diagnosis and Treatment

**DOI:** 10.3390/ijms221910347

**Published:** 2021-09-26

**Authors:** Krisztina S. Nagy, Krisztina Toth, Eva Pallinger, Angela Takacs, Laszlo Kohidai, Angela Jedlovszky-Hajdu, Domokos Mathe, Noemi Kovacs, Daniel S. Veres, Krisztian Szigeti, Kristof Molnar, Eniko Krisch, Judit E. Puskas

**Affiliations:** 1Laboratory of Nanochemistry, Department of Biophysics and Radiation Biology, Semmelweis University, Nagyvárad tér 4, 1089 Budapest, Hungary; s.nagykriszti@gmail.com (K.S.N.); toth.krisztina.105@gmail.com (K.T.); hajdu.angela@med.semmelweis-univ.hu (A.J.-H.); 2Department of Genetics, Cell- and Immunobiology, Semmelweis University, Nagyvárad tér 4, 1089 Budapest, Hungary; pallinger.eva@med.semmelweis-univ.hu (E.P.); angela.takacs1@gmail.com (A.T.); kohlasz2@gmail.com (L.K.); 3Hungarian Center of Excellence for Molecular Medicine (HCEMM), In Vivo Imaging Advanced Core Facility, Semmelweis University Site, 1094 Budapest, Hungary; mathe.domokos@med.semmelweis-univ.hu; 4Department of Biophysics and Radiation Biology, Semmelweis University, 1094 Budapest, Hungary; kovacsnoi@hotmail.com (N.K.); veres.daniel@med.semmelweis-univ.hu (D.S.V.); szigeti.krisztian@med.semmelweis-univ.hu (K.S.); 5CROmed Translational Research Centers Ltd., 1094 Budapest, Hungary; 6Department of Food, Agricultural and Biological Engineering, College of Food, Agricultural, and Environmental Sciences, The Ohio State University, 222 FABE, 1680 Madison Avenue, Wooster, OH 44691, USA; molnar.182@osu.edu (K.M.); molnarnekrisch.1@osu.edu (E.K.)

**Keywords:** folate-targeted cancer diagnosis, polymer conjugates, multivalency, chemo-enzymatic synthesis, in vitro screening, in vivo screening

## Abstract

This paper focuses on preliminary in vitro and in vivo testing of new bivalent folate-targeted PEGylated doxorubicin (DOX) made by modular chemo-enzymatic processes (FA_2_-dPEG-DOX_2_). A unique feature is the use of monodisperse PEG (dPEG). The modular approach with enzyme catalysis ensures exclusive γ-conjugation of folic acid, full conversion and selectivity, and no metal catalyst residues. Flow cytometry analysis showed that at 10 µM concentration, both free DOX and FA_2_-dPEG-DOX_2_ would be taken up by 99.9% of triple-negative breast cancer cells in 2 h. Intratumoral injection to mice seemed to delay tumor growth more than intravenous delivery. The mouse health status, food, water consumption, and behavior remained unchanged during the observation.

## 1. Introduction

The optimal approach for cancer treatment is to target exclusively cancer cells with a chemotherapeutic agent without systemic effects. The advantage of such a drug delivery system is reduced cytotoxicity and, consequently, minimized devastating side effects [1]. Much of the current cancer research focuses on the development of diagnostic and therapeutic agents that target receptors (e.g., folate receptor FR or biotin receptor BR) that are overexpressed on the surface of cancer cells. The most extensive data available in the literature discuss compounds containing folic acid (FA), targeting FR [2]. The two major groups of compounds studied are small molecule drug conjugates (SMDC) and polymeric drug conjugates (PDC).

**SMDCs**. The basic design of SMDCs is the conjugation of FA via an amide bond, a spacer to increase water solubility, and attaching the drug via chemical bonds (amide, ester, or disulfide linkages or peptide sequences) that would be cleaved in the cancer cells, releasing the drug [3,4,5]. The synthesis of these molecules is very complicated and expensive, and almost all contain only one targeting FA ligand [6,7]. Since 2000, several clinical trials with SMDCs have been aborted because they did not show the promised improvements [8,9,10,11,12,13,14].

**PDCs.** Research carried out with various polymers, including PEG, showed promise due to increased water solubility and circulation time in the body, and multivalent attachment to receptors FR [15,16,17,18,19,20]. Despite the progress achieved in the field, several obstacles currently impede the clinical translation of PDC-based therapeutics [21]. As was emphasized in a recent review, one of the greatest challenges that remain is the physicochemical heterogeneity of PDCs [21]. The uncontrolled conjugation of therapeutic agents to polymeric carriers contributes to this heterogeneity, resulting in polydisperse polymer mixtures with varied drug loadings and sites of modification. One example is poly(amidoamine) (PAMAM) dendrimer PDCs studied by the University of Michigan. It was found that a PDC with an average of ~5 folic acid (FA) targeting ligands and ~5 methotrexate (MTX) drug molecules, delivered via tail vein injections to mice with xenograft tumors, reduced tumor volume by a factor of two as compared to the equivalent dose (5.0 mg/kg) of free MTX or control saline injections [22]. The mice receiving free MTX lost all their hair and appeared sick, while those receiving MTX conjugated to the FA-PAMAM-MTX retained their hair and acted healthily. However, HPLC analysis of the dendrimer sample mentioned above with ~2.6 and 5 FA measured by ^1^H NMR showed that only 13% and 10% of the samples had 3 and 5 FA attached, ~20% had no FA, ~20% contained only 1 FA, and the rest had more than 3 or 5 ligands [23]. Thus, ~40% of the sample would have no multivalent binding.

Against this background, we set out to investigate monodisperse PDCs with precise functionalities and covalently attached drugs. This new class of PDCs is based on monodisperse functionalized poly(ethylene glycol) (“discrete” or dPEG), carrying exactly two FA-targeting groups and two doxorubicin (DOX) on each molecule, and are to be produced by patented chemo-enzymatic routes [2,24,25,26,27]. The design was based on the University of Michigan’s fundamental studies and our own investigations. Quantitative measurements of the equilibrium constant K_d_ with surface-immobilized FR receptors showed that the PAMAM conjugated with an average of 2.6 FA exhibited three orders of magnitude enhancement of binding avidities to FRs as compared to free FA [28]. Increasing the FA substitution from 2.6 to an average of 5 FA offered a much smaller improvement (one order of magnitude enhancement). The origin of the exponential change in KD was found to arise from an exponential decrease in the dissociation rate (k_off_) of the material as a function of the number of FA ligands attached. Based on this background, our group investigated fluorescein (FL)-labeled PEGs containing two FA (FA-FL-PEG-FL-FA, D), in comparison with SMDCs with one or two FA (FA-FL, and FA-FL-FA), all made by chemo-enzymatic methods with excellent yield (95%+) and selectivity (100%) [29]. FA-FL-FA had better endocytosis than FA-FL. Comparison of FA-FL-PEG-FL-FA-containing PEG with M_n_ = 1000 and 2000 g/mol and dPEG_20_ (“discreet” PEG with precisely 20 repeat units with no polydispersity) demonstrated the best uptake for dPEG in both MDA-MB-231 (Caucasian) and MDA-MB-468 (African American, less FR) triple-negative breast cancer (TNBC) cell lines [2,24]. This was the first example of using dPEG in FR-targeted PDCs. The uptake of FA-FL-PEG-FL-FA was monitored in vivo with a rat liver cancer model [30]. When the PDC was delivered intravenously, extensive tissue autofluorescence was observed with no localization in the tumor. However, local (intra-arterial) delivery resulted in accumulation in the tumor. Significant fluorescence was also observed in explanted livers in the tumor region. FL is excellent for cell culture studies, but it bleaches, and tissue fluorescence interferes with monitoring [22]. Therefore, we designed the new PDC platform using doxorubicin (DOX) as both a drug and an imaging agent. DOX was selected because it fluoresces in the red so tissue fluorescence would not interfere. Figure 1 shows the synthetic strategy.

Compounds I-III were made by “green” chemistry using *Candida anarctica* lipase B (CALB) as biocatalyst. FA-γ-SH was made by a combination of chemo-enzymatic processes, as reported [25]. In the last two steps, FA-γ-SH and DOX were attached by CALB-catalyzed Michael addition. The complete chemical structure can be seen in Appendix A in Appendix A. This paper presents the first results of in vitro and in vivo screening of this compound, focusing on the demonstration of the feasibility of in vivo monitoring of the biodistribution of the FA_2_-dPEG-DOX_2_.

## 2. Results

### 2.1. In Vitro Study

Viability studies with FA_2_-dPEG-DOX_2_ on MDA-MB-231 triple-negative breast cancer (TNBC) cell line with overexpressed FR, as demonstrated earlier [18], showed practically no cytotoxicity within 24 h incubation. In comparison, free DOX showed cytotoxic effect after 24 h already at 0.1 μM concentration (Figure 2a). After 48 h long treatment with FA_2_-dPEG-DOX_2_, the viability of the cells was reduced to 75% of the untreated control even at the lowest (0.01 μM) concentration and remained below the control level at all other concentrations applied. In comparison, the cytotoxicity of free DOX increased with increasing concentration, killing all cells at 100 μM (Figure 2b).

Long-term cytotoxicity study showed (Figure 2c) that after treatment with FA_2_-dPEG-DOX_2_ at 100 μM concentration (regarding DOX), viability values of the treated MDA-MB-231 cells were similar to the untreated control until 48 h. However, treatment with FA_2_-dPEG-DOX_2_ reduced cell viability significantly after 48 h (compared to the control) and sustained at an approximately constant level for 168 h. The results show slow DOX release in the cell culture medium investigated. Similar studies often do not investigate simulated drug release profiles because they may not be predictive for in vivo conditions [15,16,17,18,19,20].

Figure 3 demonstrates the phase-contrast morphology of the MDA-MB-231 cells treated with different concentrations of free DOX and FA_2_-dPEG-DOX_2_ for 48 h. In the case of free DOX, normal cell morphology with processes can be observed at concentrations lower than 1 μM. However, treatment with free DOX at more than 1 μM resulted in an increasing number of dead cells with a rounded shape. Regarding FA_2_-dPEG-DOX_2_ treatment, the cells retained their normal morphology (elongated cell shape with two or three processes) under 100 μM, but at this high concentration, cell death can already be seen.

Applying the Zeiss Celldiscoverer 7 automated platform, internalization of the dPEG-conjugated drug could be visualized. According to the images (Figure 4), the FA_2_-PEG-DOX_2_ formulation was able to enter into the majority of the MDA-MB-231 cells even in 5 min (Figure 4a), and it can be detected in practically every cell after 30 min (Figure 4b). Since the FA_2_-PEG-DOX_2_ was not cytotoxic in the first 24 h, it was concluded that it does not contain free DOX, only in a conjugated form.

In order to quantify total doxorubicin uptake for different drug formulations, flow cytometry was used (Figure 5). Uptake of both the free and conjugated DOX by all the cells was demonstrated. Detected fluorescence intensity values demonstrated that the drug uptake is concentration-dependent for both formulations (Figure 5a,b) since the measured intensity values at 10 µM concentration were one order of magnitude higher than in the case of 1 µM. Although the conjugated DOX has also entered all cells, the fluorescence intensity values indicate that the free drug form is taken up by the cells in slightly greater amounts than the modified drug (Figure 5c,d).

In summary, flow cytometry analysis showed that at 10 µM concentration, both DOX and FA_2_-dPEG-DOX_2_ would be taken up by 99.9% of the cells in 2 h.

FA-targeted compounds have been shown to be internalized by FR-mediated endocytosis [2,3,21,22,24]. The cellular uptake mechanism was not investigated in detail in this work, but based on Luo et al.’s work [31], we theorize that the main pathway is clathrin-mediated endocytosis due to the size of the dPEG [32], perhaps assisted by caveolae-mediated endocytosis as suggested.

### 2.2. In Vivo Testing

Preliminary testing of this compound in a live nude mouse model under isoflurane anesthesia using a two-dimensional fluorescent optical imaging showed suitable localization in induced prostate cancer (PC3-PSCA-PSMA) tumor when delivered via intratumoral injection (IT) (Figure 6c, control Figure 6a). The tumor xenograft was made using a prostate cancer cell line transfected with prostate-specific membrane antigen (PSMA) overexpression. PSMA is a folate hydrolase membrane enzyme overexpressed at the surface of different cancer cells, mostly prostate cancer, glioma but also in the neo-vasculature of breast cancers.

PSMA is the responsible enzyme for the increased uptake of FA to cancer cells. Intravenous (IV) (tail-vail vein) injection showed less uptake in the tumor and more systemic distribution, but competition with free FA uptake clearly favored the PDC (Figure 6b).

Mice were followed for 42 days post-intratumoral single injection. During Days 1, 5, 8, 12, 21, 28, 35 through 42 post-injection, three-dimensional MRI scans were obtained to account for exact tumor volumetric changes (Figure 7). We could identify intratumoral fluorescence of FA_2_-dPEG-DOX_2_ at the same excitation/emission wavelengths for 5 days after intratumoral administration.

The increase in the tumor volumes was slowed down until Day 21. After this time point, the intravenously injected mouse tumor grew in a faster manner than the tumor of the intratumorally injected animal (Figure 8.). Throughout the study, the intratumoral injection seemed to delay tumor growth more than the intravenous route of delivery. The mouse health status, food, water consumption, and behavior remained unchanged during the observation.

## 3. Discussion

Viability studies using FA_2_-dPEG-DOX_2_ on MDA-MB-231 triple-negative breast cancer (TNBC) cell line with overexpressed FR showed significant differences compared to free DOX, both administered at different concentrations (Figure 2a,b). As expected, free DOX proved to be an efficient agent in killing the MDA-MB-231 cells, whereas the PDC could only reach 75% cell viability irrespective of its concentration. Phase-contrast microscopy (Figure 3) confirmed the presence of dead cells mixed with healthy cells at 100 μM concentration. This indicates either a slow release of DOX from the PDC, thus a relatively low concentration, or bad internalization of the PDC. To find out the cause of underperformance, first, a long-term viability study was carried out. Until 48 h, FA_2_-dPEG-DOX_2_ performed similarly to the control showing no cytotoxicity to MDA-MB-231 cells. However, cell viability was reduced significantly after 48 h (compared to the control) and sustained at an approximately constant level for 168 h. Secondly, internalization of the PDC in 30 min into the majority of the MDA-MB-231 cells was confirmed by fluorescent imaging (Figure 4). The exact uptake of free DOX and the PDC was quantified by flow cytometry (Figure 5). The measurements showed that both samples have concentration-dependent uptake. DOX is taken up in slightly higher amounts that can be attributed to its smaller size compared to the PDC’s. In summary, flow cytometry analysis underlined that at 10 µM concentration, both DOX and FA_2_-dPEG-DOX_2_ would be taken up by 99.9% of the cells in 2 h. This clearly indicates that proliferation of the MDA-MB-231 cells is dramatically inhibited, possibly due to the slow release of DOX from FA_2_-dPEG-DOX_2_ instead of bad internalization.

In the presented initial proof-of-concept mouse xenograft tests, we could capture the indirect measurements of effects due to a slow release of the active cytotoxic agent DOX from the polymer. Intratumoral injection clearly made it possible for the active ingredient to reside in the tumor for a longer time than when injected intravenously. The obviously higher initial and then distribution concentration post intratumoral injection was visible in the fluorescent readings over time. The slowing growth effect of the slowly released DOX agent can also be implicated from the growth difference between the tumor of the intravenously injected animal and that of the intratumoral injection, obtained by MRI tumor volumetry measurements (Figure 7 and Figure 8).

The in vivo proof-of-concept qualitative tumor growth delay study indicated that the compound may be encouraging in regard to tumoristatic effects following both application methods, with favoring intratumoral injection. Given the initial and qualitative nature of this study on xenografted Nu/Nu mice, we could not perform a valid statistical analysis, but the large differences encountered between the sham-injected and the test compound-injected animals point to clear merit for further, more detailed, and statistically well-powered preclinical investigation of tumoristatic effects, which is currently our ongoing task. The results presented in this paper demonstrate that this methodology is feasible for a detailed study we plan for a research grant proposal. Furthermore, we are convinced that this methodology would prove successful in larger, preclinical antitumor trials.

## 4. Materials and Methods

FA_2_-dPEG-DOX_2_ was synthesized as shown in Figure 1 and was characterized by NMR spectroscopy. The details will be published elsewhere.

DOX powder-control; dimethyl-sulfoxide (DMSO) (Applichem GmbH, Darmstadt, Germany), MDA-MB-231 triple-negative human breast cancer cell line (ECACC, Salisbury, UK); Dulbecco’s Modified Eagle Medium (DMEM) (Lonza, Basel, Switzerland), Eagle’s Minimal Essential Medium (EMEM) (Lonza, Basel, Switzerland), fetal bovine serum (FBS) (Gibco, Amarillo, TX, USA), L-glutamine (Gibco, Amarillo, TX, USA), penicillin-streptomycin mix (Gibco, Amarillo, TX, USA), Hoechst 33342, Trihydrochloride, Trihydrate solution (10 mg/mL, Thermo Scientific, Waltham, MA, USA), CellTiter Glo Luminescent Cell Viability Reagent (Promega, Madison, WI, USA).

### 4.1. Cytotoxicity Study

For comparison of the cytotoxic effect of free DOX and FA_2_-dPEG-DOX_2_ on MDA-MB-231 cells, 10,000 cells/well were seeded into the white-walled optical bottom 96-well plates (Thermo Scientific, Waltham, MA, USA) in 100 μL/well of the following growth medium: DMEM (4 mg/L folic acid) was supplemented with 10% FBS, 2 mM L-glutamine, and 100 units/mL penicillin and 100 mg/mL. After 24 h, the old-growth medium was removed, and 100 μL/well of the low-folate medium was added to the cells in order to induce overexpression of folate receptors on the cell surfaces. This low-folate medium consisted of EMEM (1 mg/L folic acid) supplemented with 10% FBS, 2 mM L-glutamine, and 100 units/mL penicillin and 100 mg/mL. Following the 24 h-long incubation at 37 °C, the cells were treated with 5 different concentrations of free DOX or FA_2_-dPEG-DOX_2_ (0.01, 0.1, 1, 10, and 100 μM, regarding DOX). Firstly, DOX and FA_2_-dPEG-DOX_2_ were dissolved in DMSO at a concentration of 0.02 M (regarding DOX), which was further diluted in the completed low-folate medium. Subsequently, cell viability assay was performed 24 h and 48 h after the beginning of the treatments, applying the CellTiter Glo luminescent reagent. Luminescence was measured by means of a Fluoroskan™ FL Microplate Fluorometer and Luminometer (Thermo Scientific, Waltham, MA, USA).

Since the dPEG-conjugated DOX did not influence the cell viability appreciably during the abovementioned short-term experiment, the long-term effect of treatment with FA_2_-dPEG-DOX_2_ was also studied. For this purpose, 5000 cells/well were seeded into the white-walled optical bottom 96-well plates (Thermo Scientific, Waltham, MA, USA) in 100 μL/well of the EMEM-based low-folate medium. After 24 h, cells were treated with FA_2_-dPEG-DOX_2_ at 100 μM concentration. Firstly, DOX and FA_2_-dPEG-DOX_2_ were dissolved in DMSO at a concentration of 0.02 M (regarding DOX), which was further diluted. Dissolving and diluting of FA_2_-dPEG-DOX_2_ were carried out as it was described above. Cell viability was assessed 24, 48, 72, 144, and 168 h using the CellTiter Glo reagent, and the medium of the cells was replaced every day with fresh low-folate medium-containing supplements and FA_2_-dPEG-DOX_2_ with 100 μM DOX concentration. Luminescence was measured by a Fluoroskan™ FL Microplate Fluorometer and Luminometer (Thermo Scientific, Waltham, MA, USA).

### 4.2. Phase-Contrast and Fluorescence Microscopy

In order to observe the changes in cell morphology due to free DOX and FA_2_-dPEG-DOX_2_, phase-contrast microscopic images were taken 48 h after the beginning of the treatments with 5 different concentrations regarding DOX (from 0.01 to 100 μM) using the Axio Observer A1 inverted microscope (Zeiss, Oberkochen, Germany) and the AxioVision LE64 Rel.4.9.1. program.

To visualize the internalization process of the dPEG-conjugated doxorubicin, MDA-MB-231 cells were seeded into 24 well plates (21,000 cells/well, 10,000 cells/cm^2^) in the DMEM-based growth medium (0.5 mL/well). Following 24-h-long cultivation at standard conditions, the old-growth medium was replaced in each well with 0.5 mL of low-folate medium, and the cells were incubated for 2 h at 37 °C. Afterwards, the cell nuclei were stained with Hoechst 33342, Trihydrochloride, Trihydrate solution (10 mg/mL, Thermo Scientific, Waltham, MA, USA) diluted with DMEM to 5 μg/mL concentration. After 20 min-long incubation with the Hoechst stain, the cells were treated with FA_2_-dPEG-DOX_2_ at 16 μM concentration (regarding FA), and the 24-well plate was immediately replaced into the Zeiss Celldiscoverer 7 (Zeiss, Oberkochen, Germany) automated platform, where it was incubated for 2 h at 37 °C. After the beginning of the treatments, images were taken every 5 min for 2 h (brightfield, blue and red channels). As a result of the Hoechst staining, the cell nuclei can be seen in blue color. Due to the autofluorescent property of doxorubicin, the orange color inside the cells indicates the internalization of the drug.

### 4.3. Flow Cytometry

In order to quantify the internalization of free DOX and dPEG-conjugated DOX, fluorescence-activated cell sorting (FACS) analysis was performed. For FACS analysis, 94,000 cells were seeded into Petri dishes with 3.5 cm diameter in 2 mL of DMEM-based growth medium (10,000 cells/cm^2^). After 24 h, the old-growth medium was replaced in each Petri dish with 2 mL of low-folate medium, and the cells were incubated for 2 h at 37 °C. Then the cells were treated with 1 or 10 μM concentrations of free DOX or FA_2_-dPEG-DOX_2_ and incubated for 2 h at 37 °C. Subsequently, single-cell suspensions were prepared by using the TrypLE™ Express Enzyme solution (Gibco, Amarillo, TX, USA), washed (PBS, pH 7.4), pelleted (800 g, 5 min), and examined on a FACSCalibur flow cytometer (Becton-Dickinson Instruments, Franklin Lakes, NJ, USA). The results were evaluated with the CellQuest Pro (Becton-Dickinson Instruments, Franklin Lakes, NJ, USA) and Flowing Software 2.5.1. (Turku Bioscience Centre, Turku, Finland).

Twenty-five thousand cells (gated events) were counted for each sample, and doxorubicin fluorescence was detected with logarithmic settings on the FL2 channel (Ex 488 nm; Em 585 ± 42 nm). Untreated cells were used as the negative control.

### 4.4. In Vivo Imaging—Preliminary Studies

NMRI FOXn1nu/nu mixed-sex mice (Janvier, Bretagne, France) were used for the studies, with n = 2 control mice, n = 2 intravenously (IV)-treated mice, and n = 1 intratumorally (IT)-treated mouse. Animals had *ad libitum* access to food and water and were housed under temperature-, humidity-, and light-controlled conditions. All procedures were conducted in accordance with ARRIVE guidelines and the guidelines set by the European Communities Council Directive (86/609 EEC) and approved by the Animal Care and Use Committee of Semmelweis University (protocol number: PE/EA/599-5/2021). Mice were 30–37 weeks old with an average body weight of 34.1 ± 7.6 g. The left hind leg of mice was subcutaneously injected with 100 μL PBS containing 2 × 10^6^ prostate cancer, 3 cell line expressing prostate stem cell antigen and prostate-specific membrane antigen (PC3-PSCA-PSMA) cells, a kind gift from the Department of Immunology at Carl Gustav Carus University (Dresden, Germany), 7 weeks before the experiments. Tumor volume was monitored by weekly caliper measurements and subsequent three-dimensional magnetic resonance imaging.

250–330 µL (2.5 g/mL in DMSO-saline) FA_2_-dPEG-DOX_2_ was administered to the animals (IV or IT, depending on the mouse group) (Table 1).

### 4.5. Magnetic Resonance Imaging (MRI)

MRI measurements were performed in vivo with a nanoScan^®^ PET/MR system (Mediso, Budapest, Hungary), having a 1 T permanent magnetic field, 450 mT/m gradient system using a volume transmit/receive coil with a diameter of 60 mm. Experiments were performed with the mice under isoflurane anesthesia (5% for induction and 1.5–2% to maintain the appropriate level of anesthesia; Arrane^®^, Baxter, Newbury, UK). The T1-weighted MRI images were collected at six different time points (pre-injection, 5-, 8-, 14-, 21-, and 28-day p.i.). The MRI scans were performed with gradient echo (T1 GRE 3D) images acquired with 54 × 45 mm FOV, matrix size 110 × 92, slice thickness of 0.5 mm, 10 averages, TR/TE/FA 15/5/20, dwell time 25 ms. Images were further analyzed with Fusion (Mediso Ltd., Budapest, Hungary) and VivoQuant (inviCRO LLC, Boston, MA, USA) dedicated image analysis software by placing appropriate volume of interests (VOIs) on the tumors.

### 4.6. In Vivo Fluorescent Imaging

In vivo fluorescent imaging was performed using a two-dimensional fluorescent optical imaging instrument (FOBI, Neoscience Co. Ltd., Suwon-si, Korea). Experiments were performed in mice under isoflurane anesthesia after the MRI scans. The biodistribution images were collected at eight different time points (pre-injection, 0, 1, 5, 8, 14, 21, and 28-days post-injection) with an excitation wavelength of 480 nm corresponding to the excitation maximum of DOX (ex: 494 nm; em: 512 nm in water). The emission spectrum of the dye was in the pass band of the used emission filter. Image acquisition parameters were the following: exposure time: 2000 msec and gain: 1. The images were evaluated with VivoQuant software (Invicro-Konica Minolta, Boston, MA, USA).

### 4.7. In Vitro Absorbance and Fluorescence Tests

On the fifth day of the experiment, urine was collected from each mouse for fluorescence spectral analysis. Absorption spectra of the collected urine samples (without any modification) were recorded by a Nanodrop 1000 Spectrophotometer (Thermo Fisher Scientific, Wilmington, DE, USA) in the 220–748 nm range by 3 nm steps. Fluorescence emission spectra of the collected urine samples and the administered FA_2_-dPEG-DOX_2_ solution were recorded by Nanodrop 3300 Fluorospectrometer (Thermo Fisher Scientific, Wilmington, DE, USA) using the “white” excitation mode with 597 ± 20 nm virtual emission filter.

## Figures and Tables

**Figure 1 ijms-22-10347-f001:**
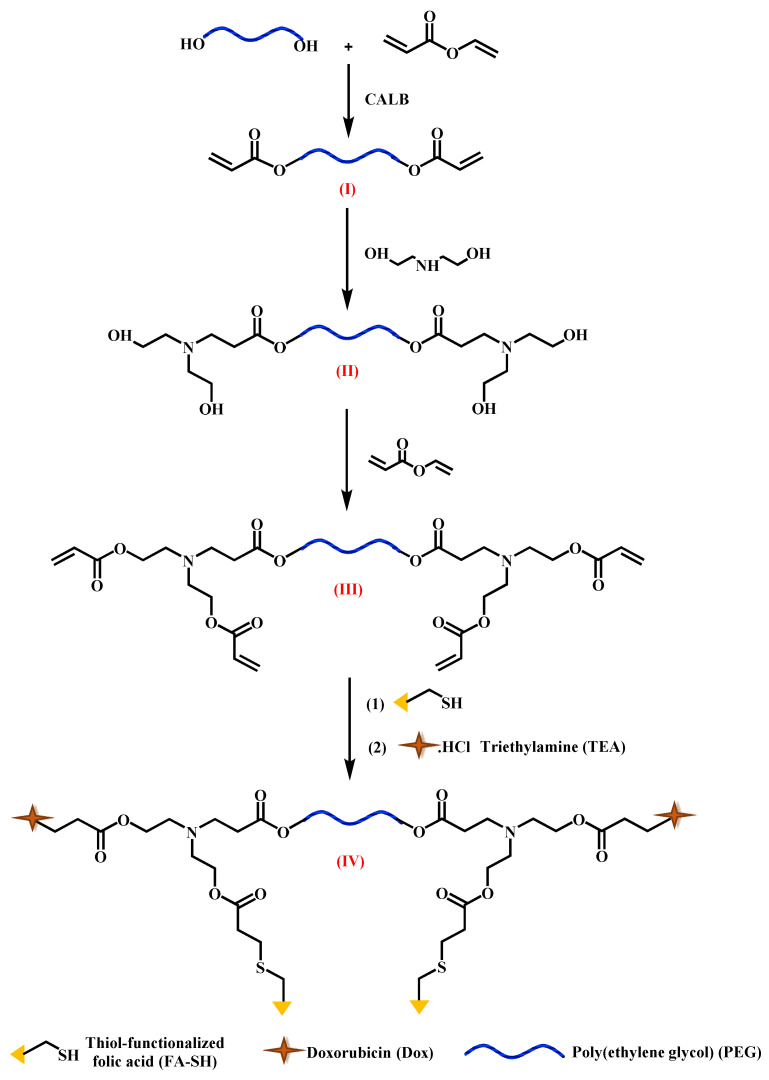
Strategy for the synthesis of FA_2_-dPEG-DOX_2._

**Figure 2 ijms-22-10347-f002:**
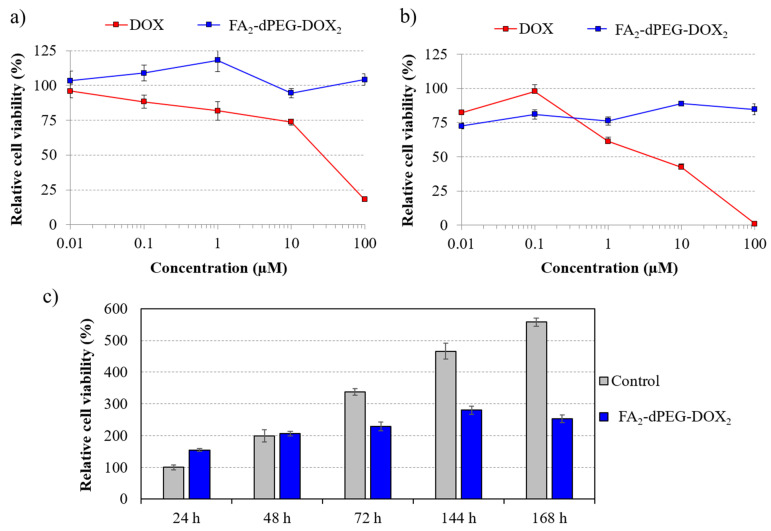
Results of the cytotoxicity assays. Relative cell viability of MDA-MB-231 cells vs. concentration of free DOX and FA_2_-dPEG-DOX_2_ after 24 h-long (**a**) and 48 h-long (**b**) treatment. Long-term effect of FA_2_-dPEG-DOX_2_ at 100 μM concentration on cell viability (**c**). All data were normalized to the viability value of untreated control measured 24 h after the beginning of the treatment.

**Figure 3 ijms-22-10347-f003:**
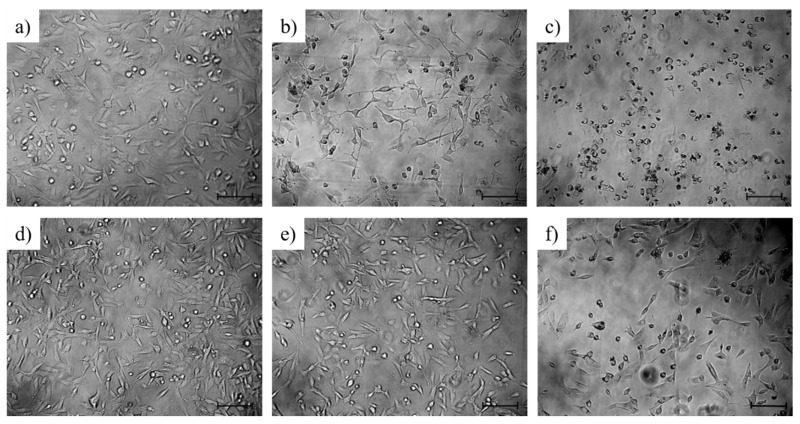
Phase-contrast microscopic morphology of MDA-MB-231 cells after 48 h-long treatment with free DOX (**a**–**c**) and FA_2_-dPEG-DOX_2_ (**d**–**f**) at concentrations of 0.01 μM (**a**,**d**), 1 μM (**b**,**e**), and 100 μM (**c**,**f**) regarding DOX. The scale bars indicate 100 μm.

**Figure 4 ijms-22-10347-f004:**
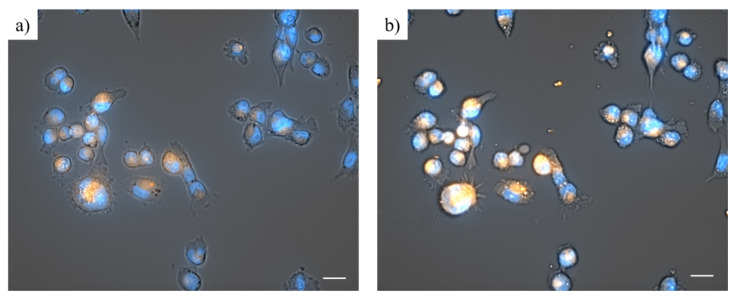
MDA-MB-231 cells 5 min (**a**) and 30 min (**b**) after the beginning of the treatment with FA_2_-dPEG-DOX_2_ at 16 μM concentration (regarding FA). The blue color shows the cell nuclei, while the orange color is the doxorubicin. The scale bars indicate 25 μM. The images were taken by the Zeiss Celldiscoverer 7 automated platform.

**Figure 5 ijms-22-10347-f005:**
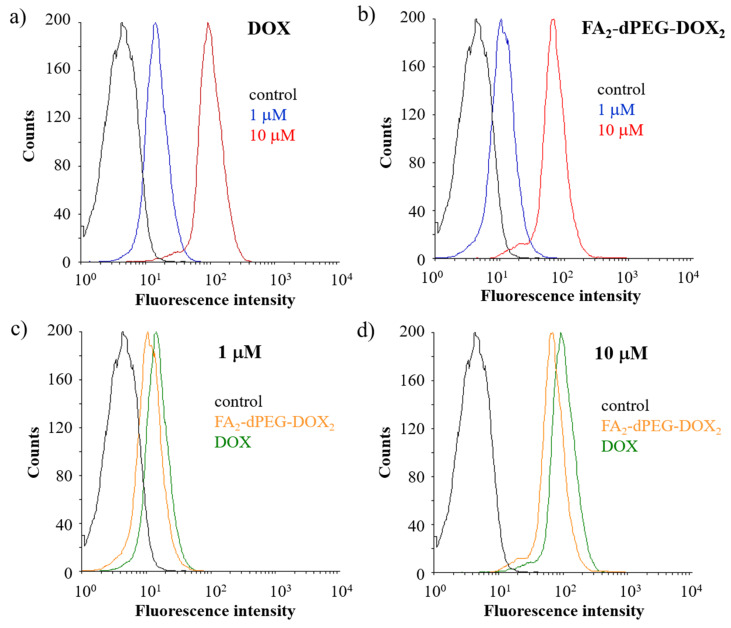
Flow cytometric measurement of doxorubicin uptake into triple-negative human breast cancer cell line (MDA-MB-231) cells. Concentration-dependent uptake of free doxorubicin (**a**) and the conjugated doxorubicin (**b**) (black line: untreated control; blue line: 1 µM; red line: 10 µM). Comparison of DOX and FA_2_-PEG-DOX_2_ internalization with representative figures regarding the uptake of free doxorubicin and the conjugated drug at a concentration of 1 µM (**c**) and 10 µM (**d**) (black line: untreated control; orange line: FA_2_-dPEG-DOX_2_; green line: free doxorubicin).

**Figure 6 ijms-22-10347-f006:**
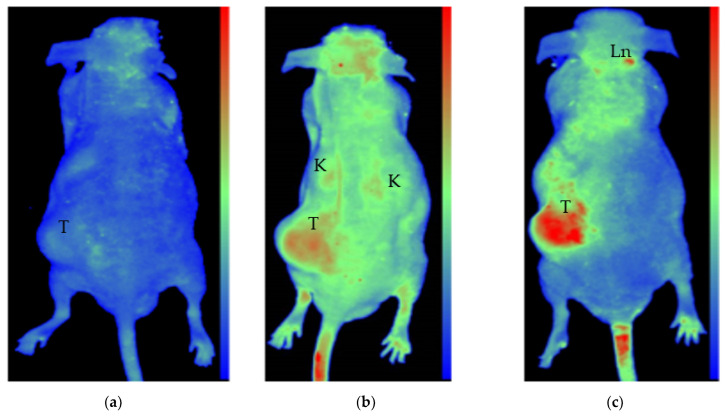
In vivo fluorescent images from the dorsal view of Foxn^Nu/Nu^ nude mice bearing LNCaP prostate xenograft tumors. Images before (**a**) and 24 h after IV (**b**) in one mouse, and after 24 h after intratumoral injection in another mouse (**c**). All images are standardized to identical light radiance minima and maxima in relative light intensity per pixel arbitrary units to allow for direct comparison. Fluorescence of the PDC is identified in the tumor (T) tail vein, kidneys (K), and capillary-rich head nuchal skin in (**b**), while a distinct fluorescent signal is observed in the tumor (T) and some lymph nodes of the neck (Ln) in the case of intratumoral PDC injected animal (**c**). The slight autofluorescence, as seen in (**a**) before injection, is clearly different from the fluorescent signals of injected animals.

**Figure 7 ijms-22-10347-f007:**
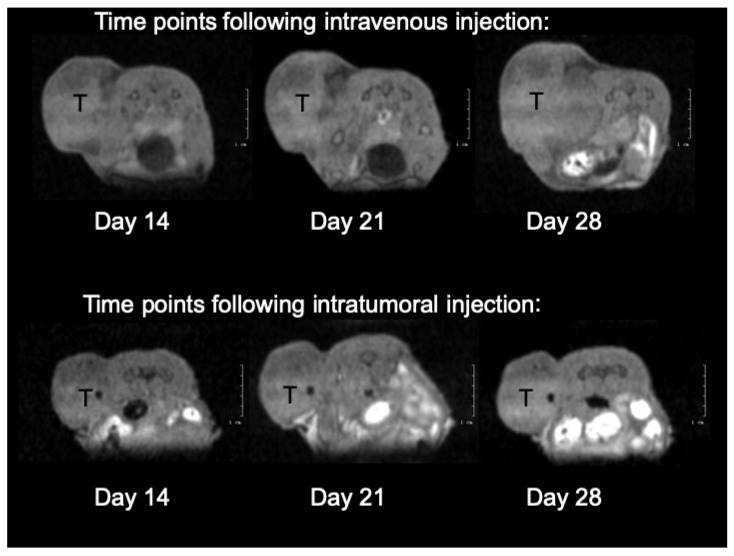
Illustration of mouse xenograft tumor growth through the presentation of three time points of MRI measurements with a cross-section through the tumor inoculation site. Tumors are denoted in their cross-section with a black T.

**Figure 8 ijms-22-10347-f008:**
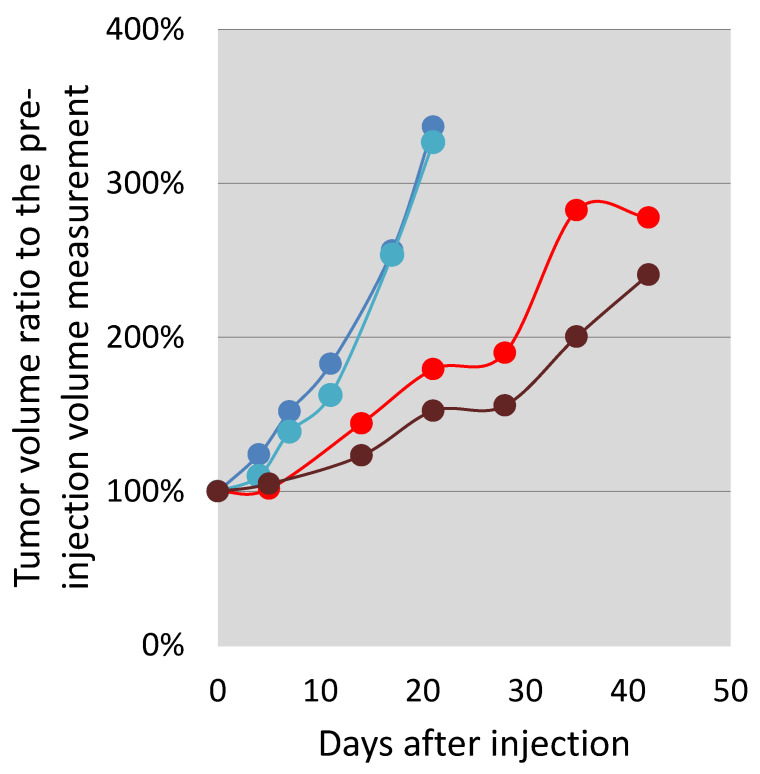
Follow-up of tumor volume increase in percentage of the initial tumor volume in a PC3-PSMA overexpressing cell line xenograft in two saline control (blue and turquoise) Nu/Nu nude mice, one injected intratumorally (brown) and one injected intravenously (red) with the PDC.

**Table 1 ijms-22-10347-t001:** The FA_2_-dPEG-DOX_2_ solution composition used in the imaging study.

Amount	Unit	Compound	APITitle 3
128.2	mg	FA2-PEG-DOX	API
50	uL	DMSO	2.50%
2000	uL	Saline	
62.5	mg/mL	Final API concentration	

## Data Availability

The data presented in this study are available on request from the corresponding author.

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
