# Peer review of "Folate-Targeted Monodisperse PEG-Based Conjugates Made by Chemo-Enzymatic Methods for Cancer Diagnosis and Treatment"

_ijms, 2021, doi:10.3390/ijms221910347_

Round 1

Reviewer 1 Report

Nagy et al. reported a new bivalent folate 24 targeted, PEGylated Doxorubicin (DOX) made by modular chemo-enzymatic processes (FA2-dPEG-25 DOX2). It is interesting, novelty, and useful. Although the authors have already stated that the data presented in the manuscript is the very preliminary result with no statistical analysis, the data presented in a published paper should be very formal and standardized. Thus, all the data in the manuscript should be added to statistical analysis.

Author Response

  1. In the title of the article the authors state:”…conjugates made by Chemo-enzymatic Methods”….However, the synthesis of the conjugate is not presented in detail. Even, the actual structure of the conjugate is not shown. In Materials and Methods the authors explain that the synthesis will be published elsewhere. In my opinion, at least the chemical structure of the conjugate must be disclosed in the present work. Furthermore, unless the synthesis implies a new methodology there is no scientific reasons to not include the synthesis in the present article.

As per the request of the Reviewer, the full chemical structure of the conjugate was added to the supplementary document as Figure S1 and was referred to at the end of the Introduction.

  1. L227 “possibly due to the slow release of DOX from FA2-dPEG-DOX2 instead of bad internalization”. It is not clear how DOX is attached to the conjugate. This detail is important to understand the mechanism of release of the drug. In addition, the release of the drug must be quantified according to such mechanism (pH, presence of proteases, lipases…).

DOX was connected to the conjugate via an amide bond that likely cleaves slowly. The full chemical structure was added to the supplementary document as Figure S1 to aid the reader. The release of DOX from the conjugate will be part of an upcoming study.

  1. In connection with DOX release: L149 “In order to quantify total doxorubicin uptake for different drug formulations, flow cytometry was used “It is not clear whether the fluorescence detected by cytometry is due to the conjugate or the free drug. Probably, the study of the drug release would shed light on this matter.

Free doxorubicin gives a fluorescent signal that can be detected by a flow cytometer. Fluorescent signal was also detected when cells were treated with the conjugated form of the drug. The same channel could be used to detect fluorescence as for the free drug. In the opinion of the Authors this means that the conjugation did not affect the excitability of doxorubicin. Because FACS can only detect cell-associated fluorescent signals, it also confirms internalization of the conjugate into the conjugate-treated cells. Therefore, when measuring the conjugate, both the free and the conjugated DOX is accounted. Since the conjugate is not cytotoxic initially, one can assume that there is no free DOX residue in this case. Therefore, the measured fluorescence in flow cytometry actually is coming from the conjugate and not free DOX.

  1. The length of the linker between the conjugate and FA is important for a proper interaction with the receptor. The nature of the linker must be assessed for an optimal performance of the conjugate.

The Authors designed the whole PDC and therefore the linker, so that it can be synthesized using the chemo-enzymatic approach. This enabled excellent selectivity and yield for many of the synthesis steps. Optimization of the linker structure in regards to interaction and internalization of the PDC into the cells was not carried out. However, based on the FACS study presented in the paper, the PDC could be easily internalized into the cells and only showed a slightly worse performance compared to free DOX.

  1. Dox is quite cationic at physiological pH, hence, the uptake of the conjugate is expected to be significand by a pasive mechanism. The mechanism of internalization should be determined by saturation experiments or at low temperatures.

Regarding internalization of free DOX, the main mechanism is indeed passive transport. However, several pieces of evidence have accumulated demonstrating that active, carrier-mediated processes are also involved by means of membrane transporters like organic anion-transporting polypeptides (OATP) or solute carrier transporters (SLCs) (Lee HH et al. 2017 Mol. Pharmacol., Kullenberg F, et al. 2021 Cells). FA-targeted compounds have been shown to be internalized by endocytosis. The new compound is much bigger than free DOX but is internalized just as fast. We will investigate this in more detail.

  1. L238: “These are very preliminary results with no statistical analysis, but the observations are very promising”. I agree. The observations are promising. But the conclusions are far from preliminary. They are, in my opinion, premature.

We wanted to demonstrate the capability of the proposed in vivo imaging, but at this point we do not have enough animals for statistical analysis. The images support our assertion that intratumoral delivery could prove to be more effective in its anti-tumoral effect. We added newly measured data for control mice to allow for a more reliable qualitative comparison of tumor growth inhibition by the compound and delivery methods presented in our manuscript.

The in vitro data has statistical analyses, so the conclusions are valid. Of course, we agree that more in vivo investigations will be needed to fully support our conclusions, but the current data are very encouraging in the regard of tumoristatic effects of the compound following both application methods, with a favor to intratumoral injection. We included this addition to the Conclusions section too.

Figure 8 has been enhanced with the measured data of two control animals, though the limited time available for the added experiments has not yet allowed the full 42-day follow-up (which, in our view, should have been terminated earlier anyway in the untreated tumor xenograft animals due to the tumor sizes reaching ethical limits for seriously hampering the normal life of animals).

To Revised Manuscript:

Figure 8. caption: Follow-up of tumor volume increase in percentage of the initial tumor volume in a PC-3-PSMA overexpressing cell line xenograft in two saline control (blue and turquoise) Nu/Nu nude mice, one injected intratumorally (brown) and one injected intravenously (red) with the PDC.

To Discussion:

The in vivo proof-of-concept qualitative tumor growth delay study indicated that the compound may be encouraging in regard of tumoristatic effects following both application methods, with a favor to intratumoral injection. Given the initial and qualitative nature of this study on xenografted Nu/Nu mice, we could not perform a valid statistical analysis, but the large differences encountered between the sham-injected and the test compound-injected animals point to a clear merit for further, more detailed and statistically well powered preclinical investigation of tumoristatic effects, which is currently our ongoing task. 

Reviewer 2 Report

SIGNIFICANCE OF THE WORK:

This article reports the in vitro and in vivo evaluation of a folate-doxorrubicin conjugate.

NOVELTY:

To the best of my knowledge the material presented is original.

COMMENTS:

  1. In the title of the article the authors state:”…conjugates made by Chemo-enzymatic Methods”….However, the synthesis of the conjugate is not presented in detail. Even, the actual structure of the conjugate is not shown. In Materials and Methods the authors explain that the synthesis will be published elsewhere. In my opinion, at least the chemical structure of the conjugate must be disclosed in the present work. Furthermore, unless the synthesis implies a new methodology there is no scientific reasons to not include the synthesis in the present article.

  1. L227 “possibly due to the slow release of DOX from FA2-dPEG-DOX2 instead of bad internalization”. It is not clear how DOX is attached to the conjugate. This detail is important to understand the mechanism of release of the drug. In addition, the release of the drug must be quantified according to such mechanism (pH, presence of proteases, lipases…).

  1. In connection with DOX release: L149 “In order to quantify total doxorubicin uptake for different drug formulations, flow cytometry was used “It is not clear whether the fluorescence detected by cytometry is due to the conjugate or the free drug. Probably, the study of the drug release would shed light on this matter.

  1. The length of the linker between the conjugate and FA is important for a proper interaction with the receptor. The nature of the linker must be assessed for an optimal performance of the conjugate.

  1. Dox is quite cationic at physiological pH, hence, the uptake of the conjugate is expected to be significand by a pasive mechanism. The mechanism of internalization should be determined by saturation experiments or at low temperatures.

  1. L238: “These are very preliminary results with no statistical analysis, but the observations are very promising”. I agree. The observations are promising. But the conclusions are far from preliminary. They are, in my opinion, premature.

Author Response

We wanted to demonstrate the capability of the proposed in vivo imaging approach, but at this point we do not have enough animals for statistical analysis. We therefore now included text into the Results section detailing that it would not be scientifically appropriate to conduct statistical analysis on the proof-of-concept experiments with such few animals, but these results are presented as the qualitative indicator towards further, statistically relevant investigations using the same in vivo approach. The in vitro data presentation does include a proper statistical analysis. We have included all details by appropriate scientific standard to allow for reproduction of our experiments.

Round 2

Reviewer 2 Report

  1. L227 “possibly due to the slow release of DOX from FA2-dPEG-DOX2 instead of bad internalization”. It is not clear how DOX is attached to the conjugate. This detail is important to understand the mechanism of release of the drug. In addition, the release of the drug must be quantified according to such mechanism (pH, presence of proteases, lipases…).

DOX was connected to the conjugate via an amide bond that likely cleaves slowly. The full chemical structure was added to the supplementary document as Figure S1 to aid the reader. The release of DOX from the conjugate will be part of an upcoming study.

Reviewer: The mechanism of release of the drug and the kinetics of such release are a fundamental aspects of any drug delivery system including conjugates. In my opinion, it must be included in the present manuscript.

  1. Dox is quite cationic at physiological pH, hence, the uptake of the conjugate is expected to be significand by a pasive mechanism. The mechanism of internalization should be determined by saturation experiments or at low temperatures.

Regarding internalization of free DOX, the main mechanism is indeed passive transport. However, several pieces of evidence have accumulated demonstrating that active, carrier-mediated processes are also involved by means of membrane transporters like organic anion-transporting polypeptides (OATP) or solute carrier transporters (SLCs) (Lee HH et al. 2017 Mol. Pharmacol., Kullenberg F, et al. 2021 Cells). FA-targeted compounds have been shown to be internalized by endocytosis. The new compound is much bigger than free DOX but is internalized just as fast. We will investigate this in more detail.

Reviewer: In the title, the authors state that a “Folate-targeted Monodisperse PEG-based Conjugates” have been prepared. Hence, it is mandatory to demonstrate whether the uptake of the conjugate is mediated by folate receptors or a pasive mechanism as in the case of DOX.

Author Response

We thank the reviewer for his comments. The following text was included in the revision: FA-targeted compounds have been shown to be internalized by FR-mediated endocytosis [2,3,21,22,24]. The cellular uptake mechanism was not investigated in detail in this work, but based on Luo et al.’s work [31] we theorize that the main pathway is clathrin-mediated endocytosis due to the size of the dPEG [32], perhaps assisted by caveolae-mediated endocytosis as suggested. We also added the following text to the Conclusion section. The in vivo proof-of-concept qualitative tumor growth delay study indicated that the compound may be encouraging in regard of tumoristatic effects following both application methods, with favoring intratumoral injection. Given the initial and qualitative nature of this study on xenografted Nu/Nu mice, we could not perform a valid statistical analysis, but the large differences encountered between the sham-injected and the test compound-injected animals point to a clear merit for further, more detailed and statistically well powered preclinical investigation of tumoristatic effects, which is currently our ongoing task. We hope that the reviewer will accept our responses.

Round 3

Reviewer 2 Report

Please, take as article model the following paper:

Synthesis and activity of a folate targeted monodisperse PEG camptothecin conjugate. Bioorganic & Medicinal Chemistry Letters 23 (2013) 5810–5813 (attached). 

In this paper the selectivity of the conjugate is assessed by addition of an excess of folate to Human KB cells in order to saturate the receptor (Figure 3) to block the uptake of the conjugate. 

Additional references: 

Journal of Controlled Release 208 (2015) 106–120 

RSC Adv., 2015, 5, 71164–71173